# Effects of Imagery Rescripting Versus Rumination on Emotion Regulation in an Online Pilot Study

**DOI:** 10.3390/bs16010059

**Published:** 2025-12-30

**Authors:** Marta Drujan, Kaltrina Gashi, Andreas J. Fallgatter, Anil Batra, Kristina Fuhr

**Affiliations:** Department of Psychiatry and Psychotherapy, Tübingen Center for Mental Health (TüCMH), University of Tübingen, Calwer Str. 14, 72076 Tübingen, Germanyandreas.fallgatter@med.uni-tuebingen.de (A.J.F.); anil.batra@med.uni-tuebingen.de (A.B.)

**Keywords:** emotion regulation, autobiographical recall, imagery rescripting, rumination

## Abstract

Imagery rescripting (IR) has been shown to be effective for emotion regulation in both patients with mental disorders and in experimental settings. However, the effects of rumination on emotion regulation remain ambiguous, with some studies classifying rumination as an adaptive emotion regulation strategy and others as maladaptive. In this pilot study, we aimed to compare the effect of IR with rumination on emotion regulation following sad autobiographical recall. We expected IR to be more effective in recovering positive affect compared to rumination and a control group. In an online experiment, 70 healthy students were randomly assigned to IR, rumination, or a passive control group after recalling a sad autobiographical memory. Mood was assessed using the Positive and Negative Affect Schedule (PANAS) before and after the intervention. In this study, IR was superior to rumination (*p* = 0.033) in recovering PA after sad autobiographical recall. Therefore, IR affects mood recovery positively after recalling a sad autobiographical memory. Rumination resulted in prolonged impairment in mood and therefore should be regarded as a maladaptive emotion regulation strategy.

## 1. Introduction

Over the last decades, imaginative methods have been widely adopted across various traditions of psychotherapy and developed for everyday clinical practice ([7]). One such method is imagery rescripting (IR) ([2]; [10]). IR is a technique through which people are instructed to recall a past aversive event and give it a better outcome in their imagination ([24]). IR has its roots in the work of Pierre Janet and Fritz Perls, as well as in early hypnotherapy, and it has been integrated into cognitive behavioral therapy and schema therapy ([7]; [2]; [48]). The use of IR by various schools of psychotherapy in the past, and the lasting usage of IR in the present, give rise to the reasonable assumption that this method may be based on universal mechanisms of action, such as emotion regulation ([38]). Imagery is thought to be particularly important for emotional regulation, as images evoke stronger emotions than words ([13]). Cognitive techniques like reappraisal or reframing focus on changing the perspective on a certain (negative) situation verbally by changing attitudes and thoughts. Meanwhile, IR achieves change through experienced emotions and images of the recalled memory. Furthermore, some laboratory studies indicate that IR affects negative and positive emotions more strongly than verbal processing ([12]; [14]). Aversive mental imagery plays a crucial role in a variety of mental disorders. Conclusively, it makes sense to target aversive mental memories imaginatively ([13]). For this reason, IR is particularly used for the treatment of recurrent aversive images in PTSD ([1]; [16]; [37]) or social phobia ([21]). Moreover, studies suggest its efficacy in the treatment of personality disorders, social anxiety disorder, agoraphobia, depression, and eating disorders ([24]). However, although IR is widely used and the underlying mechanism of action is already being discussed for depression ([11]; [46]), a comprehensive conclusion about the mechanism of action has yet to be drawn ([36]; [38]). Overall, there are only a few studies to date that experimentally investigate the effects of imaginative interventions on emotion regulation.

To better understand emotion regulation, studies have asked participants to recall negative autobiographical memories to induce a negative mood of personal relevance. This paradigm was first described in 1982 ([47]). Because aversive memories and childhood experiences play a crucial role in addressing both depression and anxiety disorders within psychotherapeutic work (see also Beck’s model, e.g., [3]), autobiographical paradigms are advantageous for studying emotion regulation through imaginative intervention. Other studies using autobiographical memories in laboratory settings have shown that IR can reduce the emotionality associated with the memory ([36]; [38]). [38]’s ([38]) research further reinforces that IR reduces the frequency of aversive autobiographical memories and increases feelings of mastery. In adolescents, IR of a recalled bullying situation led to a reduction in negative affect (NA) ([44]). [6] ([6]) compared cognitive restructuring techniques with IR in people with bulimia nervosa who recalled an aversive memory of a social rejection situation. Both cognitive and imaginative techniques were equally effective in improving the management of negative emotions, leading to an improvement in positive emotions and a reduction in negative emotions. These laboratory studies on IR for aversive autobiographical memories provide evidence that IR may work through emotion regulation. Previous studies that investigated IR were conducted with a clinical population ([24]; [6]). However, studies addressing the underlying emotion regulation processes in a laboratory setting chose a student sample ([47]; [38]) or a community sample ([36]).

The importance of internet-based interventions has increased, particularly during the COVID-19 pandemic. A review article published in 2021 suggests that internet-based IR is no less effective than face-to-face interventions ([27]). An IR intervention for self-help in depression successfully reduced depressive symptoms and increased self-esteem and quality of life ([25]). Eating disorder prevention with IR has also been investigated in an online format ([29]). IR improved body satisfaction and increased compassion for oneself compared to a cognitive intervention. In conclusion, IR may be an effective emotion regulation strategy that can be successfully applied in an online format. To date, no study has investigated whether IR can restore NA induced by recalling a negative autobiographical situation in an online format.

In a previously conducted experimental laboratory study, we compared an ‘inner safe place’ intervention to a rumination task through the participants’ changes in positive (PA) and negative affect (NA). For this purpose, firstly, a negative mood was induced in all participants using autobiographical recall. Subsequently, half of the participants were presented with a version of the ‘inner safe place’ and the other half with a rumination task based on [15] ([15]) via headphones. With a total of 54 participants, as expected, the ‘inner safe place’ was superior to rumination in terms of improving positive affect (PA) but not NA after negative mood induction ([5]). Thus, we have already shown that the ‘inner safe place’ imagery intervention can effectively increase PA. In addition, high imagination capacity (measured with the Tellegen Absorption Scale, TAS) significantly predicted mood improvement in PA.

### Aims of the Study

The aim of the present study was to investigate the potential emotion regulatory effect of IR versus self-reflection (rumination) on a negative autobiographical event in a non-clinical sample of young adults. This study was designed as a pilot study for the acquisition of preliminary data before a larger study is conducted. The larger study will compare the effects of the inner safe place with IR and rumination in a clinical population. The IR intervention was based on [25] ([25]), who used IR in a sample of depressed patients. Their IR script was adapted into a tape-recorded version. As there was no control group other than rumination in the previous study ([5]), a passive control group with no specific task was added as an additional control condition. Based on previous emotion regulation research, we expected that after recalling a sad autobiographical event, healthy participants in the IR group would experience a greater improvement in PA than participants in a rumination or passive control group. We also wanted to investigate whether imagination capacity (absorption) influences mood changes after IR or rumination. We did not expect effects on NA in accordance with previous research ([5]).

## 2. Materials and Methods

### 2.1. Study Design

This study was designed as a mixed 3 × 3 factorial experiment. The first factor was time, measured at baseline (t0), after autobiographical recall (t1), and after the intervention (t2). The second factor was the type of intervention: imagery rescripting (IR) versus rumination (RUM) versus a passive control group (CONT). The study was approved by the local ethics committee (214/2022BO2) and conducted entirely online via the platform SoSciSurvey ([22]).

### 2.2. Participants

Subjects were recruited from the University of Tübingen and the University of Saarland via emails announcing the study to mentally healthy students and members of the university (age ≥ 18 years). The study was advertised as a study investigating how participants cope with sad emotions. We informed participants that they would be randomized to either the intervention or one of the two control conditions. However, we did not give any details about the type of intervention being studied. As compensation, participants were offered the chance to participate in a prize drawing to win one of five EUR 15 vouchers for a local bookstore.

A total of 70 people participated in the study (58 women, 11 men, 1 diverse). The mean age was 27.86 years (*SD* = 8.54, range 18–50). All participants were fluent in German, mentally healthy (they confirmed via self-disclosure before participation that they were not currently suffering from depression, anxiety disorder, bipolar disorder, schizophrenic/psychotic disorder, an eating disorder, or borderline personality disorder), and provided written informed consent for study participation. Using the online platform SoSciSurvey, 27 participants were randomly assigned (by SoSciSurvey) to IR, 23 participants were assigned to RUM, and 20 participants were assigned to CONT. The characteristics of the participants are shown in Table 1. There were no significant group differences (all *p* > 0.146).

### 2.3. Materials

#### 2.3.1. Mood Assessment

Current mood was assessed using the German version ([19]) of the Positive and Negative Affect Schedule (PANAS) ([43]) at baseline (t0), after autobiographical recall (t1), and after IR vs. RUM vs. CONT (t2). To avoid sequential effects, all 20 items were presented in a randomized order implemented via SoSciSurvey. The instrument has been shown to be reliable and has been used in various studies that also target emotion regulation or use imagery interventions ([15]; [33]).

#### 2.3.2. Absorption

The German version ([34]) of the Tellegen Absorption Scale (TAS) ([39]) was used to assess openness to absorption and self-altering experiences, a construct that covers aspects such as hypnotic suggestibility and the tendency of participants to become highly involved in imaginative experiences, fantasy, and memories, as well as also synesthesia and dissociation (see [35]; [41]).

#### 2.3.3. Autobiographical Recall

The autobiographical recall task was adapted from [47] ([47]) and accompanied by a recommended piece of music, ‘Adagio for Strings’ by Samuel Barber. This procedure has been used in previous studies to induce a sad mood ([9]; [33]; [45]). The instructions were tape-recorded and presented simultaneously with the music over headphones for four minutes. Participants were asked to recall a past sad event in their life and remember as many details as possible (e.g., “Who was with you at this event?”, “What was the environment like?”) while the music played in the background. Subjects were advised to perform the task thoroughly.

#### 2.3.4. Imagery Rescripting

In the IR instructions, participants were asked to reflect on their aversive autobiographical memory by modifying their recalled autobiographical situation. The imagery-based modification was conducted by the participant until they perceived their revised memory positively. The modification could involve an external superhero or internal superpowers. The instructions were adapted from a previous study by [25] ([25]) and are available online on the authors’ website ([26]). Participants listened to the IR instructions through headphones. The audio recording lasted six minutes in total.

#### 2.3.5. Rumination Task

Based on [15]’s ([15]) procedure, participants were instructed to focus on their thoughts, feelings, and bodily expressions related to the memory of their autobiographical recall, e.g., “Think about how your body feels right now.”, “Think about what your feelings might mean”. There were 28 items with a 20 s pause between to allow the participant to think about the items, with the aim of inducing rumination. In total, the recorded rumination task lasted six minutes.

#### 2.3.6. Control Group

In CONT, participants were instructed to wait for five minutes. No further instructions were given for or during this period. At the end of the experiment, participants were asked about the activities they in engaged in during this period. About ten participants in CONT reported that they were waiting, thinking, resting, or reflecting on the experiment or a personal situation. Nine participants reported distracting themselves via reading messages or a book, watching television, browsing the internet, or doing house chores.

#### 2.3.7. Control Questions

After completing the experiment, all participants were asked how well they were able to perform the intervention (using a visual analogue scale of 0–100) and how the intervention was experienced. Given the open-ended responses, their extensive length, and the lack of resources for qualitative analyses, the details of the feedback on the experiment are not reported.

### 2.4. Procedure

The participants first gave their written informed consent online, answered demographic questions, and completed the TAS and PANAS (t0). They then recalled the aversive autobiographical memory and completed the PANAS (t1). Subsequentially, they either listened to the IRinstruction, the RUMinstruction, or were assigned to CONT. The final part included completing the PANAS (t2) and answering some control questions. Participants who were assigned to CONT had the opportunity to listen to the IR after the experiment was completed. After completion, participants were debriefed. In total, the experiment took 20 to 30 min.

### 2.5. Statistical Analysis

Analyses were conducted and figures were created using the statistics programs R ([32]) and SPSS Statistics Version 28.0 with one-tailed tests and an alpha level of *α* = 0.05. To analyze the primary hypothesis, i.e., the effect of the different conditions on PANAS-PA and PANAS-NA scores, separate repeated measures analyses of variance (RM-ANOVAs) were conducted with the PANAS-PA or PANAS-NA scores as the dependent variable, condition (IR vs. RUM vs. CONT) as the independent between-subjects variable, and assessment time (t1 vs. t2) as the independent within-subjects variable. We tested for an interaction between time and condition and used post hoc Bonferroni comparisons between conditions. An a priori sample size calculation using G-Power ([8]) for the primary hypothesis indicated a power of 0.95 with a total sample size of 66 subjects for an estimated effect size of *f* = 0.25 and a correlation between repeated measures of 0.5. The design and analysis plan of this study were not pre-registered, but they were pre-approved by the local ethics committee.

We tested the association between age, gender (dummy coded: 1 = male and 2 = female; the non-binary person was excluded for statistical reasons), and absorption with the change in PA and NA using Pearson correlations. The change was calculated between t1 (after recall) and t2 (after intervention) in PA and NA (t2 − t1). ANCOVAs were then calculated with the significant covariates (or ANOVAS with a new factor for dummy-coded variables).

## 3. Results

### 3.1. Effect of Aversive Autobiographical Recall on Positive and Negative Affect

As shown in other studies, there was a main effect of time for PA, *F*(1,67) = 104.23, *p* < 0.001, and NA, *F*(1,67) = 44.14, *p* < 0.001, indicating a significant change from t0 (before) to t1 (after recall). The impairment of PA after autobiographical recall did not differ between the experimental conditions for PA, *F*(2,67) = 0.29, *p* = 0.746. NA increased less in the RUM condition during recall compared to the other conditions, *F*(2,67) = 3.58, *p* = 0.033 (see also Table 2). There were no main effects of the condition for PA, *F*(2,67) = 0.92, *p* = 0.403, or NA, *F*(1,67) = 0.59, *p* = 0.559.

### 3.2. Effect of Imagery Rescripting on Positive and Negative Affect

Means and standard deviations of PANAS scores at t1 and t2, as well as the change scores, are displayed in Table 2. Both PA and NA scores of the three conditions during all assessments (including baseline) are shown in Figure 1.

Regarding PA, RM-ANOVA showed a significant effect of time *F*(1,67) = 16.85, *p* < 0.001, indicating a significant improvement after sad mood recall in all conditions. There was an expected interaction between time and condition *F*(2,67) = 3.49, *p* = 0.036, and a significant main effect for condition, *F*(2,67) = 3.48, *p* = 0.037. Participants after rumination showed less improvement and less PA compared to those after IR (post hoc Bonferroni *p* = 0.033) but did not differ from CONT (post hoc Bonferroni *p* = 0.317). IR and CONT did not differ in the post hoc test (Bonferroni *p* > 0.999).

RM-ANOVA for NA resulted in a significant effect of time *F*(1,67) = 36.56, *p* < 0.001, but no main effect for condition, *F*(2,67) = 1.60, *p* = 0.210, and no interaction between time and condition *F*(2,67) = 1.68, *p* = 0.195. NA decreased after all conditions.

### 3.3. Relationship Between Absorption and Mood Change

There was a small significant correlation between change in PA and absorption (measured with the TAS), *r* = 0.24, *p* = 0.046, but not with age (*r* = −0.07, *p* = 0.588) or gender (*r* = 0.10, *p* = 0.414, *n* = 69). No association between change in NA and absorption (*r* = −0.15, *p* = 0.202), age (*r* = 0.16, *p* = 0.200), and gender (*r* = −0.16, *p* = 0.181, *n* = 69) can be reported.

In the ANCOVA with the TAS as a covariate, the main effect of condition was reduced (*F*(2,66) = 3.08, *p* = 0.053) compared to an ANOVA without the TAS as a covariate (*F*(2,67) = 3.49, *p* = 0.036). The TAS, however, showed no significant main effect on PA change, *F*(1,67) = 3.32, *p* = 0.073.

## 4. Discussion

The hypothesis that the technique of IR evokes positive emotions after recalling an aversive autobiographical event was confirmed. This finding points to the underlying mechanism—namely, emotion regulation—to explain how imaginative interventions modify negative emotions in healthy adults. This result is in line with previous research on imaginative interventions to restore positive affect in experimental settings ([30]; [33]). In contrast to previous research, however, this pilot study induced sad mood with autobiographical memories but in an experimental rather than a clinical setting ([24]; [38]). Compared to our previous work ([5]), a passive control group was included in addition to the rumination task. In opposition to some of the literature, e.g., [20] ([20]), who attribute a positive effect of rumination comparable to mindfulness, the results of the present study suggest that rumination has negative effects on positive affect and appears to be a maladaptive rather than an adaptive emotion regulation strategy ([42]). In other words, focusing on negative aspects of past events inhibits a resourceful reappraisal of the past. Although this study examined healthy participants, it is likely that participants who focused on past aversive memory experienced similar negative cognitive and behavioral consequences on par with depressed patients. In particular, thinking about a potentially different outcome if occurrences had been aligned differently is both present in patients with depression and in adults who engage in rumination ([31]). Rumination is an emotion regulation strategy that is particularly prevalent in patients with depression. Impaired cognitive inhibition or control is associated with dysfunctional rumination as a maladaptive emotion regulation strategy in patients with depression ([17]; [18]).

The result that absorption/imaginative capacity was specifically associated with improvement in positive rather than negative emotions is partly consistent with the literature. For example, absorption is correlated with vividness of mental imagery and with PA, and participants with high scores on absorption scales may show a different state of consciousness similar to hypnotic induction ([28]). Hypnotic suggestibility itself is also associated with positive rather than negative outcomes in several medical conditions, such as pain ([23]). In contrast, there is also evidence of an association between hypnotic suggestibility and negative aspects, such as dissociation and dissociative psychopathology ([40]).

### Limitations

This study had some limitations to mention. First, as this was an experimental design and a study with mentally healthy participants, the results cannot be generalized to other settings or (clinical) samples. Even if the finding that rumination is a maladaptive emotion regulation strategy is in line with previous clinical research, no conclusions can be drawn for patients with depression or other mental disorders. Also, the participants’ mental health was not confirmed in a clinical or diagnostical interview. Second, the study was conducted as a pilot experiment to induce negative mood and investigate the effects on mood of small and brief interventions/emotion regulation strategies lasting no more than six minutes. IR and its use in this study, therefore, cannot be considered a clinical or psychotherapeutic intervention. Nevertheless, it can be concluded that even in a small experimental setting, IR can show effects in restoring positive mood. Third, no long-term effects were assessed after the end of the experiment, so the positive effect of IR and the negative effect of rumination on emotions may only be short-term. However, a study by [4] ([4]) indicates long-term, positive effects of IR on affect and self-esteem, which were still measurable one week after the intervention. In addition, the experiment was implemented online during the COVID-19 pandemic using an anonymized internet-based platform, and therefore inclusion and exclusion criteria (especially regarding mental health) could not be verified by assessing diagnostic criteria with clinical interviews. Finally, only a self-reported measure was used to assess the outcome.

## 5. Conclusions

IR positively affected mood recovery after recalling an aversive autobiographical memory in a mentally healthy sample of students and young adults. The finding highlights the importance of imaginative interventions not only in clinical but also in non-clinical settings. Using an online format, the study showed that participants were able to independently imagine different and more positive outcomes of a personal aversive memory. This can be interpreted as a reason for implementing IR as part of internet-based self-help interventions. Rumination, on the other hand, seems to prolong impairments in affective states and should therefore be considered a maladaptive emotion regulation strategy also in non-clinical samples. Further research is needed to confirm the conclusions about IR as an adaptive, and rumination as a maladaptive, emotion regulation strategy.

## Figures and Tables

**Figure 1 behavsci-16-00059-f001:**
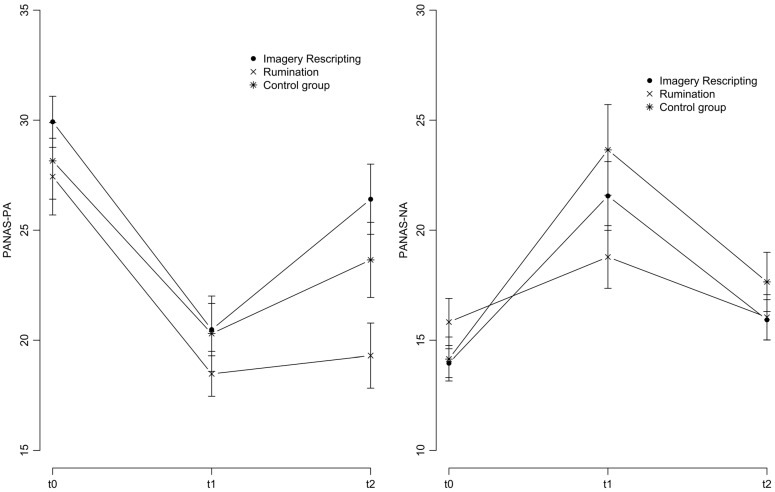
Positive and negative affect at baseline (t0), after autobiographical recall (t1), and after intervention (t2).

**Table 1 behavsci-16-00059-t001:** Sample characteristics.

	IR (n = 27)	RUM (n = 23)	CONT (n = 20)	Total (n = 70)
	n (%)	n (%)	n (%)	n (%)
Female	21 (77.8%)	20 (87.0%)	17 (85.0%)	58 (82.9%)
Male	6 (22.2%)	2 (8.7%)	3 (15.0%)	11 (15.7%)
Diverse	0 (0.0%)	1 (4.3%)	0 (0.0%)	1 (1.4%)
	M (SD)	M (SD)	M (SD)	M (SD)
Age	26.30 (7.35)	30.09 (10.43)	27.40 (7.43)	27.86 (8.54)
TAS	60.93 (19.15)	54.83 (25.04)	63.55 (23.79)	59.67 (22.51)
PANAS-PA	29.93 (6.03)	27.43 (8.36)	28.15 (7.78)	28.60 (7.34)
PANAS-NA	13.96 (3.38)	15.83 (5.14)	14.15 (4.45)	14.63 (4.35)

Notes: TAS = Tellegen Absorption Scale; PANAS-PA = Positive and Negative Affect Schedule—positive affect; PANAS-NA = Positive and Negative Affect Schedule—negative affect; IR = imagery rescripting; RUM = rumination; CONT = control group.

**Table 2 behavsci-16-00059-t002:** Positive and negative affect before and after intervention.

		IR (n = 27)	RUM (n = 23)	CONT (n = 20)	Total (n = 70)
PA	t1	20.48 (6.19)	18.48 (4.88)	20.30 (7.64)	19.77 (6.24)
	t2	26.41 (8.29)	19.30 (7.07)	23.65 (7.63)	23.29 (8.19)
	change	5.93 (7.41)	0.83 (5.76)	3.35 (7.08)	3.51 (7.05)
NA	t1	21.56 (8.12)	18.78 (6.84)	23.65 (9.21)	21.24 (8.17)
	t2	15.93 (4.77)	16.04 (4.93)	17.65 (6.02)	16.46 (5.19)
	change	−5.63 (7.30)	−2.74 (4.67)	−6.00 (7.38)	−4.79 (6.64)

Notes: Means and standard deviations are displayed; PA = Positive and Negative Affect Schedule—positive affect; NA = Positive and Negative Affect Schedule—negative affect; t1 = before intervention; t2 = after intervention; change = t2 − t1; IR = imagery rescripting; RUM = rumination; CONT = control group.

## Data Availability

All materials, data, and analyses of the study are available upon request from the corresponding author.

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
