# Peer review of "Effects of Imagery Rescripting Versus Rumination on Emotion Regulation in an Online Pilot Study"

_behavsci, 2025, doi:10.3390/bs16010059_

Round 1
Reviewer 1 Report
Comments and Suggestions for Authors
Thank you for allowing me to review this interesting manuscript on a study comparing IR, RUM and a CG via an online experiment. Given the rise of internet-based interventions, this is a trending topic. I have a few comments and suggestions.
Abstract
Please refine the aim of this study by saying that you compare IR to rumination. As of now, this is not quite clear.
Introduction
l.95, l. 100 & l. 103: Please replace "imagery rescirpting" with IR, as you already used the abbreviation. I've noticed this throughout the manuscript. Please change accordingly.
Methods
ll.124-126: Am I right that psychological disorders were assessed via self-report? Did the participants undergo an interview, or did they provide written information on possible diagnosis? You only mentioned "current diagnosis"; did you ask for previous episodes of mental illness?
Table 1: Please correct the spelling to "Diverse" or non-binary. It would also be helpful to use coherent labels, as you described gender as non-binary in the written paragraph, while here it is labelled as diverse.
ll.160f: I think this is part of the Results and should be described there.
l.192: Please indicate why you decided against reporting the feedback. Was it too long or did you not have the resources to analyze it?
l.202: Please add a space between "to30".
l.207: I think you mean the different conditions rather than the safe place induction, or did you also do this? I think this might be from your previous study?
Results
In the first paragraph, the use of abbreviations or long forms seems a bit all over the place. Please harmonize the use of abbreviations throughout the mansucript.
Figure 1: In the figure, the control group is called waiting control group. Again, please decide on one label and use this throughout the manuscript.
Discussion:
l. 260: I think the conclusion that your experiment explains the underlying mechanism that also works in e.g., bulimia nervosa and PTSD is a bit of a stretch - while this might be the case, your sample consisted of healthy participants. It might be a hint, but I feel like your conclusion is too strong here. You also point this out yourself in the Limitations section.
All in all, this is a very interesting study, and I think the main points for improvement lie in the narration, e.g., the incoherent use of abbreviations and labels.
Reviewer 2 Report
Comments and Suggestions for Authors
This paper addresses the gains from ‘imagery rescripting’ for emotion regulation, in patients with rumination. The study involved a sadness-induction autobiographical task, comparing three groups (including a rumination and a control group, totalling 70 healthy students, randomly assigned). All were assessed using the well-established PANAS measure. As expected, imagery was superior to rumination, so the effects were not enormous, and the study suggests that rumination be explicitly considered as maladaptive emotion regulation strategy (which seems sensible).
The paper is well written. The study appears to have been well-conducted, based on appropriate ethics oversight, and the data are analysed in appropriate way.
I have a few broad suggestions.
- I struggled to compare ‘rescripting’ with the related phenomena of reappraisal, reframing, or counterfactual thinking (the word reappraisal only appears once, and counterfactual thinking twice, and all only in the Discussion section). Can we view these as related phenomena, in which case, there is a large literature on reappraisal that was not reviewed. Alternatively, it would be helpful if the Introduction showed the way in which (say) rescripting and reappraisal are differentiable?
- Clearly the ‘rescripting’ approach has been widely used in the literature. However, I feel that the Introduction would benefit from framing of the decision to use this particular technique with this particular group. Is rescripting the most widely used technique, and the industry standard? Is it the most effective technique, best suited to a student population? And is this because of ease of implementation, or consistency of post-intervention continuation? Answer to these is yes. And I thought that the Introduction could do with a better justification for why this particular approach was chosen.
- The fact that the study is based on healthy students randomly assigned has, of course, some limitations. Most notably, because this is not a clinical population, the range of variation in mental health is likely to be narrow. It would be helpful if the Method section justified why the student population was chosen, and whether this group of young well-educated clinically asymptomatic people are the target population? Alternatively, is this to be viewed as a pilot study, before being rolled out with a more representative sample of the population in general? I know this is discussed in the Limitation section, but it would be helpful to have some text towards the end of the Introduction.
- While I am a big fan of the use of control conditions in psychotherapy studies, this did mean that the number of participants in each of the groups was small, and circa N=20. This meant of course that the effect sizes are modest, and it is harder to generate matched groups (for example, as regards age). The Limitation section did discuss a number of critical limitations, such as clinical applicability, brevity of intervention, and the absence of a longer-term outcome measure. However, I’m concerned that the study joins a long list of other underpowered psychotherapy investigations. This is why justifying the work (under point 2 above) is important. I also think that it is important to label the study as some sort of ‘pilot’ (including using this word in the Abstract and probably the Title), to avoid any conclusion that it should be compared with a large randomised control trial.
- I am also concerned about the evaluation of mental health status of members of these groups. Clearly, they were screened to avoid the more serious mental health conditions. However, had a subsection of one group been randomly assigned to people who were for example, mildly depressed, then this could well have skewed the results. Perhaps I missed this, but I struggled to find a measure of the mental health status of the group before the intervention, and an evaluation to show that this is matched across the groups.
In sum, I think this is a reasonable study, well-conducted and analysed and written up. However, I do think that it needs clarity about the nature of the intervention (how is re-scripting different from reappraisal different from counterfactual thinking), and a justification for the specific technique and choice of experimental subjects.
However, despite my reservations, I would be prepared to consider a revision of the paper, rather than rejection on the basis of under-powered sample size.
Round 2
Reviewer 2 Report
Comments and Suggestions for Authors
I was pleased to see that the authors appear to have addressed all of the concerns that I raised in their initial submission. This was especially important in respect to identifying the study as a 'pilot', given the sample size and demographic restrictions.
On this basis, I'm happy to accept the paper for publication.